# Uro-pathogens: Multidrug resistance and associated factors of community-acquired UTI among HIV patients attending antiretroviral therapy in Dessie Comprehensive Specialized Hospital, Northeast Ethiopia

**Mihret Tilahun**[1]\*, **Mesfin Fiseha**[1], **Mihreteab Alebachew**[1], **Alemu Gedefie**[1], **Endris Ebrahim**[1], **Melkam Tesfaye**[1], **Melaku Ashagrie Belete**[1], **Abdurahaman Seid**[1], **Daniel Gebretsadik**[1], **Ermiyas Alemayehu**[1], **Wondmagegn Demsiss**[1], **Bekele Sharew**[1], **Agumas Shibabaw**[1], **Habtamu Mekonnen**[2], **Tewodros Dessie**[2]

**1** Department of Medical Laboratory Sciences, College of Medicine and Health Science, Wollo University, Dessie, Ethiopia, **2** Amhara Public Health Institute Dessie Branch, Dessie, Ethiopia

\* tilahunmihret21@gmail.com

## Abstract

### Background

Urinary tract infections are common bacterial and fungal infections in humans, occurring both in the community and in immunocompromised patients in healthcare settings. Urinary tract infections have a significant health impact on HIV-infected patients. Nowadays, drug-resistant pathogens are widespread poses a serious clinical risk, and causes urinary tract infection. The common agents of bacteria and fungi that cause urinary tract infection are *Escherichia coli* followed by *Klebsiella pneumonia*, *Staphylococcus saprophyticus*, *Enterococcus faecalis*, *group B streptococcus*, *Proteus mirabilis*, *Pseudomonas aeruginosa*, *Staphylococcus aureus* and *Candida. albicans*. This study aimed to investigate uro-pathogen, multidrug resistance pattern of bacteria, and associated factors of community-acquired urinary tract infection among HIV-positive patients attending antiretroviral therapy in Dessie comprehensive specialized hospital, Northeast Ethiopia from February 1, 2021, to March 30, 2021.

### Methods

An institutional-based cross-sectional study was conducted at Dessie Comprehensive Specialized Hospital. Socio-demographic and clinical data were collected by using structured questionnaires from HIV patients suspected of community-acquired urinary tract infections. About 10 ml of clean-catch midstream urine was collected and inoculated into Blood agar, MacConkey, and Cysteine lactose electrolyte deficient media. Yeasts were identified by using Gram stain, germ tube test, carbohydrate fermentation, assimilation tests, and chromogenic medium. Gram stain and biochemical tests were performed to identify isolates and an antimicrobial susceptibility pattern was performed on disc diffusion techniques. Data

## Methods

An institutional-based cross-sectional study was conducted at Dessie Comprehensive Specialized Hospital. Socio-demographic and clinical data were collected by using structured questionnaires from HIV patients suspected of community-acquired urinary tract infections. About 10 ml of clean-catch midstream urine was collected and inoculated into Blood agar, MacConkey, and Cysteine lactose electrolyte deficient media. Yeasts were identified by using Gram stain, germ tube test, carbohydrate fermentation, assimilation tests, and chromogenic medium. Gram stain and biochemical tests were performed to identify isolates and an antimicrobial susceptibility pattern was performed on disc diffusion techniques. Data

**Data Availability Statement:** All relevant data are within the paper.

**Funding:** The authors received no specific funding for this work.

**Competing interests:** The authors declare that they have no competing interest.

**Abbreviations:** AIDS, Acquired immune deficiency syndrome; ARTC, Attending antiretroviral therapy clinic; ART, Antiretroviral therapy; ATCC, American Type Culture collections; BA, Blood agar; CFU, Colony Forming Unit; CLED, Cysteine Lactose Electrolyte Deficient; CONS, Coagulase-negative Staphylococci; HAART, Highly active antiretroviral therapy; HIV, Human Immunodeficiency Virus; MACA, MacConkey agar; MDR, Multi-Drug Resistant; MSU, Midstream urine; OIS, Opportunistic infections; UTI, Urinary Tract Infection; VCT, Voluntary counseling, and Testing.

were entered and analyzed using SPSS version 25. Both bivariate and multivariable logistic regression analysis was performed and a P value of < 0.05 with an adjusted odds ratio with their 95% confidence interval (CI) was used as statistically significant associations.

## Results

From the total 346 study participants, 92 (26.6%) were culture positive 75 (81.52%) were bacterial and 17 (18.48%) were fungal pathogens. From a total of 75 bacteria isolates 51 (68%) were Gram-negative bacteria and the most commonly isolated bacteria were *E. coli* 16 (21.33%) followed by *K. pneumoniae* 11(14.67%) and *enterococcus species* 10(10.87. Of the 17 fungal isolates of fungi, 8(47.1%) were represented *by C. tropicalis*. Of the isolated bacteria, 61(81.3%) were resistant to three and above classes of antibiotics (drug classes). About 13 (81.3%) of *E. coli*, 9(81.8%) of *K. pneumoniae*, 8(80%) of *Enterococcus* species, 7 (77.8%) of *P. aeruginosa*, and *CoNs* 7(87.5%) were the most frequently exhibited three and above classes of antibiotics (multi-drug resistance). Amikacin and gentamicin were effective against Gram-negative Uro-pathogens. Participants aged>44year, female, being daily labor, being farmer, unable to read and write, patients with CD4 count of $\leq$ 200 cells/mm$^3$ and CD4 count of 201–350 cells/mm$^3$, who had chronic diabetics, patients having a history of hospitalization and who had urgency of urinations were statistically significant association with significant urinary tract infections.

## Conclusion

The burden of community-acquired urinary tract infections among HIV patients is alarmingly increased. Therefore, behavior change communications might be considered for promoting the health status of HIV patients. Moreover, CD4 level monitoring and therapeutics selection based on microbiological culture are quite advisable for the management of urinary tract infections of HIV patients.

## Introduction

Urinary tract infection (UTI) is an infection of the urinary tract, including the kidneys, ureters and bladder, urethra, and accessory structures that collect, store, and release urine from the body. Urinary tract infection occurs when microorganisms, typically bacteria from the digestive tract, enter the urethral opening and begin to multiply [1–3].

Globally, the burden of UTI is estimated to be 150 million cases every year [4]. Based on previous epidemiological studies from Tanzania and elsewhere in sub-Saharan Africa, urinary tract infections were estimated from 35–45% of cases [5]. In Ethiopia, a systematic review and meta-analysis indicated that the pooled prevalence of urinary tract infection was 15.97% [6] and urinary tract infections in HIV patients in Ethiopia were 12.8% [7].

Urinary tract infections are caused by bacteria, as well as certain fungi and *Escherichia coli (E. coli)* is the most common cause of both uncomplicated and complicated UTIs, followed by *Klebsiella pneumonia (K.pneumoniae)*, *Staphylococcus saprophyticus (S.saprophyticus)*, *Entero-coccus faecalis (E. faecalis)*, *group B streptococcus(GBS)*, *Proteus mirabilis (P.mirabilis)*, *Pseudo-monas aeruginosa (P. aeruginosa)*, *Staphylococcus aureus (S.aureus)*, and *Candida* spp [8]. Evidence showed that *Escherichia coli* is the most common causative agent for complicated

UTIs. The order of prevalence for causative agents is *Enterococcus* spp., *K. pneumoniae*, *Candida* spp., *S. aureus*, *P. mirabilis*, *P. aeruginosa*, and GBS [9]. Candidauria is the presence of *Candida spp*. in the urine, such as *C. albicans*. Patients with cystitis, epididymorchitis, prostatitis, pyelonephritis, and renal candidiasis experience symptomatic candidiasis [10]. C. albicans is one of the most common fungi that can cause Candidauria (20% of nosocomial infections) [11].

In developing countries, the emergence of antibiotic resistance in the treatment of urinary tract infections is a major public health issue [12]. Recently, the most common pathogens of UTIs developed resistance to most antimicrobial agents [13]. *E. coli*, *Klebsiella species*, and *Proteus species* had high resistance rates to amoxicillin and tetracycline of 80.1%–90.0% and low sensitivity rates of 0–25% to nitrofurantoin, ciprofloxacin, and gentamicin. Different evidence across the world and Africa including Ethiopia showed that more than 50% of uropathogenic bacteria developed multidrug resistance (MDR). About 70.0% of bacteria from India [14], and Ethiopia 79.3% from Wolayta Sodo Ethiopia [15], 78.4% from Addis Ababa [16], and 91.7% from Gondar [17] developed MDR. The antimicrobial test analysis showed that 152 (47.85%) of the isolates were resistant to two or more antimicrobials [18]. The majority of treatments are initiated empirically, knowledge of the organisms, their epidemiological characteristics, adequate treatment, control of these conditions necessitated, and antibacterial susceptibility is required [19].

Urinary tract infections cause increased mortality, disability, hospitalization, economic loss in the HIV population, renal failure, and hypertension. It is predisposed to nephritic complications such as urinary tract obstruction, which causes renal colic, pelvic-ureteric junction obstruction, benign prostate hyperplasia, urethral strictures, bladder dysfunction, and urinary stasis [20].

A systematic review indicated that being female, having a history of recurrent UTI, having low socioeconomic status, having diabetes mellitus, having genitourinary tract abnormalities, having a low CD4 count, history of catheterization, history of hospitalization, and having chronic diabetic status were independent factors for the occurrence of urinary tract infection [7, 15]. Another study on factors associated with urinary tract infections among HIV-infected patients with low CD4 count, chronic hepatitis, and being female [21].

Despite numerous studies on bacterial UTI, antimicrobial susceptibility testing, and HIV infection-related factors in developed countries, there is little data on the burden of bacterial UTI and antimicrobial susceptibility patterns among UTI-symptomatic HIV-infected patients in resource-limited settings, including Ethiopia in general, and not enough study was done in the study area in particular. As a result, the purpose of this study was to determine the Uropathogens, multidrug resistance pattern of bacteria, and associated factors of community-acquired urinary tract infection among HIV-positive patients attending antiretroviral therapy (ART) clinics in Northeast Ethiopia.

## Methods and materials

### Study design, area, and period

A hospital-based cross-sectional study was conducted at Dessie Comprehensive Specialized Hospital (DCSH) from February 1, 2021, to March 30, 2021. According to the Dessie town health administrative office, in the town, there are 16 governmental health institutions (1 comprehensive specialized hospital, 1 general hospital, 8 health centers) 3 private general hospitals, 6 higher private clinics, and the hospital serves about 12 million people living in the catchment areas. The Dessie Comprehensive Specialized Hospital serves about 12 million people living in the catchment areas and has a 600 average daily patient flow also has 800 staff, among whom

about 600 are health professionals. It has 600 beds, 8 wards, and different OPDs like adult, pediatric, emergency, and TB & HIV. Also, the hospital provides different services like laboratory, pharmacy, imaging (X-ray, ultrasound, and city scan), and Dessie comprehensive specialized hospital ART service serves 9,500 HIV positive.

**Population.** The source of the population was all HIV-positive individuals attending DCSH. Whereas, HIV- positive patients who had symptomatic community-acquired UTI and visited the ART clinic of Dessie Comprehensive Specialized Hospital during the study period before 48 hours of admissions were the target populations.

**Eligibility criteria of study participants.** HIV-positive patients who were ≥18 years old with suggestive of UTI including; lower abdominal or flank pain, dysuria and hematuria, and frequency urinations were included in the study while patients who were mentally ill, patients who received antibiotics in the last 14 days before sample collection and unable to give samples were not included in the study.

**Variables.** The dependent variable the prevalence of bacterial and fungal community-acquired UTI was the dependent variable. While socio-demographic characteristics like (age, sex, marital status, educational level, residence) and clinical profile like (history of catheterization in the past year, current ART status, diabetes, CD4+cell count, cotrimoxazole usage, and history of hospitalization) in the past year were independent variables.

## Sample size and sampling technique

The required sample size was determined by using a single population proportion formula by considering the proportion of 28.6% [22] from a previous Ethiopian study, marginal error of 5%, and 95% confidence interval = 1.96 by using the following formula:

$$n = \frac{(Z_{a/2})^2 * p * (1-p)}{d^2}$$

The calculated sample size was 314. Including a 10% non-response rate a total of **346 UTI** symptomatic HIV patients who fulfilled the inclusion criteria were included consecutively.

**Operational definitions. UTI:** Urinary tract infections are one of the most common infections in humans that can happen anywhere along the urinary tract and it is caused by microorganisms (microscopic organisms that are too small to be seen by the naked eye). There are several different types of microorganisms which include bacteria, protozoa, fungi, and algae [23, 24].

**Community-acquired urinary tract infection.** (CA-UTI); Community-acquired UTI is an infection if an individual with UTI is suspected before hospital admission and specimens are collected from the outpatient or within less than 48 hours of hospital admission [25].

*History of catheterization.* The use or insertion of a catheter into the bladder or participants who have been catheterized before data collection [26].

*History of hospitalization.* Participants were taken to a health facility and kept there for 48 hours for treatment before data collection in the past six months or more months.

*Multidrug resistance.* A bacterium that is simultaneously resistant to three or more antimicrobial categories [27].

**Data and specimen collection.** A pre-tested structured questionnaire was used to collect information on socio-demographic data (age, residence, and occupation), and clinical history (existing antibiotic treatment, previous antibiotic therapy, previous history of hospitalization for a longer period). The participants' recent CD4+ count was retrieved from their medical records. About 10 mL of voided clean-catch midstream fresh urine was collected from each study participant by using a leak-proof and sterile wide-mouthed screw-capped container.

These specimens were labeled and stored in a cold box (4°C) and transported to the Wollo University College of Medicine and Health Sciences microbiology laboratory for analysis within 1 hour [28].

**Isolation and identification of bacteria.** Bacterial isolation and phenotypic characterization were performed using the recommended culture and biochemical tests [29]. A calibrated loop that delivers 0.001 mL of urine was used to inoculate each urine sample onto the Cysteine lactose electrolyte deficient agar (Oxoid Ltd, UK). The plates were incubated aerobically at 37°C for 24 hours and colony count growth of $\geq 10^4$-$10^5$CFU/mL (colony-forming units per milliliter) was considered significant. Gram stain was performed from significant growth and sub cultured onto MacConkey agar (HiMediaTM) and 5% blood agar plates (HiMediaTM). However, bacteria that did not show growth after 24 hrs of incubation were further incubated for 24 hours and discarded as negative and the colony count was not significant [30]. Colony characteristics, Gram reactions, and a series of biochemical reactions, including catalase, coagulase, oxidase, urease, indole, citrate utilization, lysine decarboxylase, glucose, lactose fermentation, gas and H2S production, and motility tests were used for the isolations of bacteria [31].

**Fungal identification.** Yeasts were identified by using routine diagnostic methods such as Gram stain, germ tube test, carbohydrate fermentation, and assimilation tests, as well as chromogenic medium (CHROMagar Candida Medium (Biomerieux, France) according to the manufacturer's instructions [32].

**Antimicrobial susceptibility testing.** An antimicrobial susceptibility test was performed by using the Kirby–Bauer disk diffusion method based on the Clinical Laboratory Standards Institute (CLSI) recommendation [33]. About 3–5 pure colonies of isolated species from nutrient agar were picked and transferred to a tube containing 5 mL of tryptone-soya broth and mixed well to make a homogenous suspension. The suspension was incubated at 37°C until the turbidity of the suspension matched a 0.5 McFarland standard. Using a sterile swab, the suspension was inoculated over the entire surface of the Mueller Hinton agar plate. The selected antimicrobial disks were put on the inoculated plates and incubated at 37°C for 16–18 hours. Antimicrobial agents were selected based on CLSI recommendations and local (Ethiopian) prescription habits for bacteria. The antimicrobials (Oxoid Ltd) that were used for bacterial susceptibility testing were amoxicillin-clavulanic acid (AMC) 10μg, ampicillin (AMP) 10μg, amikacin (Amk) 30μg, cefotaxime (CTX) 30μg, ceftriaxone (CRO) 30μg, trimethoprim-sulphamethoxazole (SXT) 25μg, ciprofloxacin (CIP) 5μg, gentamycin (Gen) 10 μg, ceftazidime (CAZ) 30μg, nitrofurantoin (F) 300μg, tetracycline (TE) 30μg and penicillin (Pen) 10μg [33]. Within 15 minutes after the application of the discs, the plates were incubated at 35°c for 18 hours. Diameters of zones of inhibition were measured using a digital caliper. The antimicrobial susceptibility test results were interpreted as sensitive, intermediate, or resistant based on the standardized CLSI guidelines and the isolates were considered MDR, resistant to at least one antimicrobial in three or more antimicrobial categories [33].

**Quality assurance.** The questionnaire was pretested on 5% (for 18) of HIV-positive patients at Dessie Health Center. The sterility of culture media was checked by incubating 5% of the prepared media overnight at 37°C without specimen inoculation. The collected data was checked for completeness and adequate recording on the worksheet both during and after data collection. Standard Operating Procedures (SOP) were strictly followed for each microbiological procedure. All clinical specimens were collected, transported, and processed correctly. The expiration dates of the media, reagents, and Muller Hilton agar antimicrobial discs were checked before use. The new batch culture medium and antimicrobial disks were checked for performance and quality using the American Type Culture Collection (ATCC) reference strains such as *E. coli* (ATCC® 25922), *S. aureus* (ATCC® 25923), *Klebsiella pneumonia* (ATCC 700603) and *P. aeruginosa* (ATCC® 27853).

**Data management and statistical analysis.** The data was entered into epi-data version 4.6.0.4 and exported to Social Sciences Statistical Package (SPSS) version 25 for analysis. Descriptive statistics were computed and presented using graphs and tables. A bivariate analysis was conducted to classify factors that separately affect the frequency of dependent variables. Variables with a P-value less than or equal to 0.25 in the bivariate analysis were subjected to multi-variable analysis. Finally, variables with an adjusted odd ratio and a p-value of <0.05 were considered statistically significant.

**Ethical considerations.** Ethical clearance and permissions were obtained from the ethical review committee of Wollo University College of Medicine and Health Sciences with protocol number CMHS 365/2021. A support letter was written to Dessie Comprehensive Specialized Hospital. Permission was obtained from Dessie Comprehensive Specialized Hospitals. Written consent was also obtained from family and surrogates to obtain information from those who can't give data. Information obtained from the study participants was kept confidential. Positive results were communicated and given to attending physicians for appropriate treatment.

## Results

### Socio-demographic characteristics

In this study, a total of 346 HIV-positive patients were included. The age of the study participants ranged from 21 to 80 years with a mean (±SD) age of 39.44 (±10.87) years, and 116 (33.5%) of them were in the age range of 35–44 years. About 52.02% (180/346) of the respondents were females. Regarding educational status, 33.2% (115/346) of the respondents had a primary level of education. Moreover, 26.01% (90/346) were self-employed and 62.1% (215/346) of them were urban dwellers (**Table 1**).

### Clinical characteristics of study participants

The clinical picture of study participants revealed that 150 (43.4%) and 140(40.5%) had a history of previous hospitalization and catheterization, of which 35(30.4%) and 42(30%) were culture-positive for community-acquired urinary tract infections, respectively. About one hundred and twenty-one (35%) of the respondents had diabetes during the period of the study, with a positive growth of 32(25.6%). About 105 (30.3%) HIV patients had 201–350 cells/mm3 CD4 count with positive growth of 35(33.3%). Moreover, 210(60.7%), 200 (57.8%), 165 (47.7%), and 164 (47.4%) participants showed dysuria, frequency of urination, flank pain, and abnormal urine color with positive growth of 58(27.6%), 55(27.5%), 61 (37%), and 64 (39%), respectively (**Table 2**).

### Prevalence of community-acquired UTI and isolated pathogens

Among 346 HIV patients, 92 (26.6%) were clinically confirmed to have UTI and of these bacteria were isolated from 75 clinically confirmed UTI patients and 17 were fungal. Among 75 bacterial isolates, the most frequent bacterium was E. *coli* which accounted for 21.33% (16/75) followed by *K. pneumoniae*, *enterococcus species*, and *P. aeruginosa*, accounting for 14.67% (11/75), 13.33% (10/75), and 12% (9/75), respectively. The least number of bacterial isolates were *Enterobacter cloacae*, which accounted for 5.33% (4 /75). Gram-positive bacteria accounted for only 32% (24/75) of the bacterial isolates, of which 41.67% (10/24) of the isolates were *Enterococcus species*. Of 17 candida isolates, 47.1% (8/17) were represented by *C. tropicalis* (**Fig 1**).

**Table 1. Sociodemographic characteristics of UTI symptomatic HIV positive patients attending Dessie Comprehensive Specialized Hospital, Dessie, Ethiopia.**

| Variable | Category | Urinary tract infection | | Total, No (%) | Percentage |
|---|---|---|---|---|---|
| | | Positive No (%) | Negative No (%) | | |
| Sex | Male | 42(25.3) | 124(74.7) | 166 | 47.98 |
| | Female | 50(27.8) | 130(72.2) | 180 | 52.02 |
| Residence | Rural | 62(28.8) | 153(71.2) | 215 | 62.1 |
| | Urban | 30(22.9) | 101(77.1) | 131 | 37.9 |
| Age (in years) | 18–24 | 10(18.9) | 43(81.1) | 53 | 15.3 |
| | 25–34 | 21(24.7) | 64(75.3) | 85 | 24.6) |
| | 35–44 | 30(25.9) | 86(74.1) | 116 | 33.5 |
| | >44 | 31(33.7) | 61(66.3) | 92 | 26.6 |
| Educational status | Unable to read and write | 42(36.5) | 73(63.5) | 115 | 33.2 |
| | Primary | 20(23.3) | 66(76.7) | 86 | 24.9 |
| | Secondary | 22(24.2) | 69(75.8) | 91 | 26.3 |
| | College/University | 8(14.8) | 46(85.2) | 54 | 15.6 |
| Marital status | Married | 45(37.2) | 76(62.8) | 121 | 35 |
| | Unmarried | 18(22.5) | 62(77.5) | 80 | 23.1 |
| | Divorced | 13(13.7) | 82(86.3) | 95 | 27.5 |
| | Widowed | 16(32) | 34(68) | 50 | 14.5 |
| Occupation | House Wife | 15(25.9) | 43(74.1) | 58 | 16.8 |
| | Civil Servant | 10(22.2) | 35(77.8) | 45 | 13 |
| | Self-Employee | 9(18) | 41(82) | 50 | 14.5 |
| | Driver | 10(20.8) | 38(79.2) | 48 | 13.9 |
| | Daily Labor | 23(25.6) | 67(74.4) | 90 | 26.01 |
| | Farmer | 25(45.5) | 30(55.5) | 55 | 15.9 |

### Antimicrobial susceptibility pattern of bacterial Uro-pathogens

**Gram negative bacteria.** From the tested antimicrobial susceptibility patterns, about 52.9%, 52.9%, and 58.8% of the total isolated Gram-negative Uro-pathogens were susceptible to ceftriaxone, ciprofloxacin, and gentamicin, respectively, whereas 51%, 72.5%, 72.5%, 84.3%, 60.8% and 60.8% of the isolates were resistant to amikacin, ampicillin, amoxicillin-clavulanic acid, trimethoprim-sulphsmethsoxazole, nitrofurantoin, and ceftazidime, respectively. About 62.5% of *E. coli* strains were sensitive to gentamycin. But, 87.5%, 87.5%, 81.3%, 75%, 62.5%, and 56% of bacterial isolates *E. coli* were resistant to tetracycline, Trimethoprim-sulpha-methoxazole, amoxicillin-Clavulanic acid, ampicillin, nitrofurantoin, and ceftazidime, respectively. Similarly, 80% of ampicillin, 90.9% of trimethoprim-sulphamethoxazole, 90.9% of tetracycline 63.6% of nitrofurantoin, and 63.6% ceftazidime, 54.5% was resistance to K. *pneumoniae*. However, (54.5%) of gentamicin, and 54.5% of both amikacin and ciprofloxacillin, were sensitive to *K. pneumoniae* (**Table 3**).

**Gram positive bacteria.** From the total isolated Gram-positive Uro-pathogens, 66.7% of isolates were susceptible to clindamycin. Whereas, 87.5%, 83.3%, 75%, 58.3%, and 58.3% of Gram-positive isolates were resistant to tetracycline, trimethoprim-sulfamethoxazole, penicillin, chloramphenicol, and nitrofurantoin, respectively. *Enterococcus species* were sensitive for 70% and 60% of clindamycin ciprofloxacin, respectively. However, *Enterococcus species* showed more resistance to 90% for tetracycline, 80% for trimethoprim-sulphamethoxazole, and 70% for nitrofurantoin. Likewise, *coagulase-negative staphylococci* (CoNs) were resistant to most of the antimicrobials 87.5% for tetracycline, 87.5% for trimethoprim-sulphamethoxazole, and 75% for penicillin and chloramphenicol each. About 50% and 62.5% of the isolates *S. aureus* and CoNS were methicillin-resistant (**Table 4**).

**Table 2. Frequency of clinical characteristics of UTI symptomatic HIV positive patients attending Dessie Comprehensive Specialized Hospital, Dessie, Ethiopia (February 1, 2021- march 30, 2021).**

| Variables | Category | Culture Result | | Frequency | Percentage |
|---|---|---|---|---|---|
| | | Positive No (%) | Negative No (%) | | |
| History of hospitalization | Yes | 35(30.4) | 115(69.6) | 150 | 43.4 |
| | No | 57(29.1) | 139(70.9) | 196 | 56.6 |
| Use of cotrimoxazole as prophylaxis | Yes | 40(26) | 114(74) | 154 | 44.5 |
| | No | 54(29.1) | 138(70.9) | 192 | 55.5 |
| Diabetes | Yes | 32(25.6) | 89(74.4) | 121 | 35 |
| | No | 60(24.6) | 184(75.4) | 244 | 65 |
| History of catheterization | Yes | 42(30) | 120(70) | 140 | 40.5 |
| | No | 50(24.3) | 136(75.7) | 206 | 59.5 |
| UTI symptoms | | | | | |
| Dysuria | Yes | 58(27.6) | 152(72.4) | 210 | 60.7 |
| | No | 34(25) | 102(75) | 136 | 39.3 |
| Frequency of urination | Yes | 55(27.5) | 155(72.5) | 200 | 57.8 |
| | No | 37(25.3) | 109(74.7) | 146 | 42.2 |
| Urine color | abnormal | 64(39) | 100(61) | 164 | 47.4 |
| | Normal | 28(15.4) | 154(84.6) | 182 | 52.6 |
| Flank pain | Yes | 61(37) | 104(63) | 165 | 47.7 |
| | No | 31(17.1) | 157(82.9) | 181 | 52.8 |
| CD4+ count | $\leq$200 cells/mm$^3$ | 24(32.9) | 49(67.1) | 73 | 21.1 |
| | 201–350 cells/mm3 | 35(33.3) | 70(66.7) | 105 | 30.3 |
| | 351–500 cells/mm3 | 20(23.8) | 64(762) | 84 | 24.3 |
| | $\geq$500 cells/mm3 | 13(15.3) | 72(84.7) | 85 | 24.6 |

## Multi-drug resistance patterns of the bacterial isolates

Overall, 70 (93.3%) bacterial isolates were resistant to at least one antimicrobial agent, and 63 (84%) isolates were resistant to $\geq$2 antimicrobials. About 11 (14.7%) isolates had resistance to five or more antimicrobials. The overall prevalence of MDR bacteria (a bacterium that is simultaneously resistant to three or more antimicrobial categories) was 81.3% (61/75). About *13* (81.3%) of *E. coli*, 9(81.8%) of *K. pneumoniae*, *8*(80%) of *Enterococcus species*, *7 (77.8%) of P. aeruginosa*, and *CoNs* 7(87.5%) were the most frequently exhibited MDR (**Table 5**).

## Factors associated with CA-UTI

In the current study, bivariate analysis was performed on sociodemographic characteristics such as sex, family size, residence, occupations, immunization status, educational status, current marital status, current CD4 count, history of hospitalizations, catheterizations, and status of diabetics; and signs and symptoms of urinary tract infections. Multivariate analysis was done by including variables that had a p-value of $\leq$0.25 in bivariate analysis. After adjustment for age, sex, occupation, educational status, current CD4+ count (cells/mm3), history of previous hospitalization and diabetes status, history of catheterizations, and urgency of the urinary tract. Age >44year [AOR = 1.56, 95%CI: 1.11–9.35, P = 0.001], being female [AOR = 3.29, 95%CI: 1.26–13.41, P = 0.002], being daily labor [AOR = 4.54, 95%CI: 1.41–21.71, P = 0.003], being farmer [AOR = 6.54, 95%CI: 1.21–23.51, P = 0.004], participants unable to read and write [AOR = 4.58, 95%CI: 1.21–13.58, P = 0.001], HIV patients with CD4 count of $\leq$ 200 cells/mm$^3$ (AOR: 7.44; 95%CI: 1.64–27.21, P = 0.001) and CD4 count of 201–350 cells/mm$^3$ (AOR: 4.55; 95%CI: 1.36–18.28, P = 0.009), patients who had chronic diabetics (AOR: 4.80;

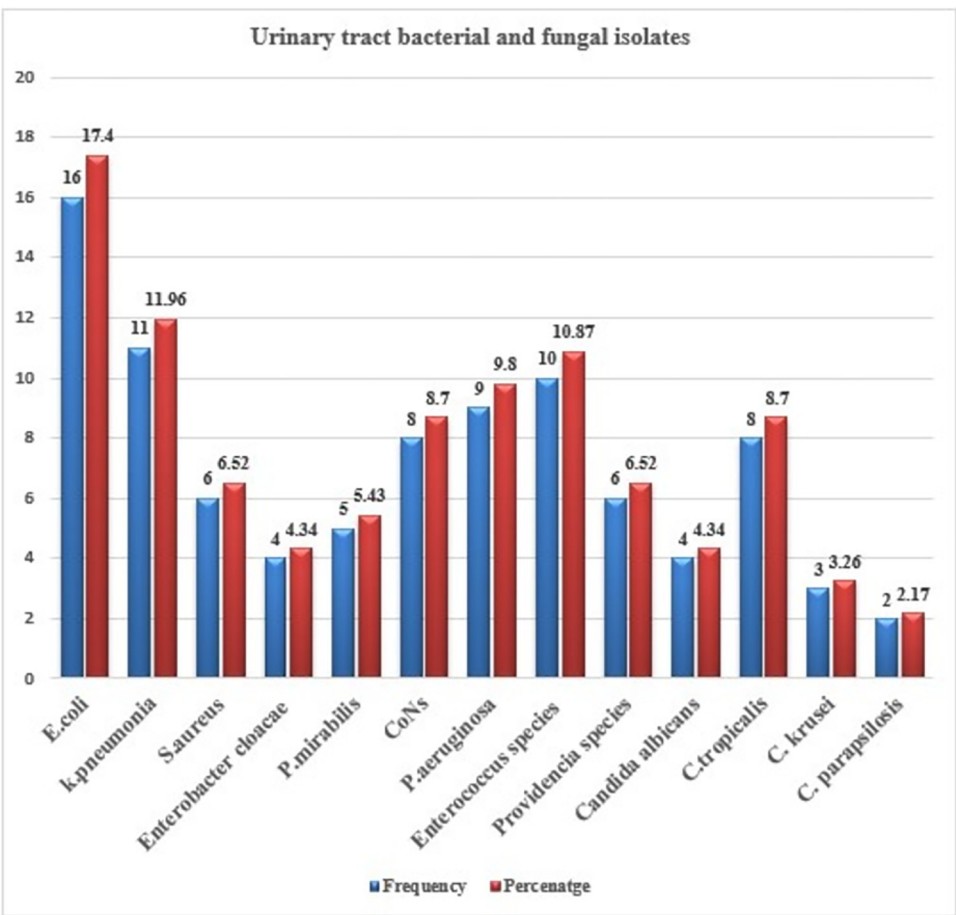

**Fig 1. Etiology of UTI among HIV patients attending antiretroviral therapy clinics of Northeast Ethiopia.**

95%CI: 1.85–16.31, p = 0.009), patients having a history of hospitalization (AOR: 6.27; 95%CI: 1.25–15.17, p = 0.002) and patients who had the urgency of urinations (AOR: 6.51; 95%CI: 1.75–36.31, p = 0.0025) were statistically significant associated risk factors for community acquired urinary tract infections (Table 6).

## Discussion

A urinary tract infection (UTI) is defined as the invasion of the urinary tract by one or more uropathogenic bacteria species, which results in significant bacteriuria and the presence of suggestive UTI such as dysuria, pain, and burning during urinating, cloudy urine, and urine appears red, bright pink [34]. In this study, the prevalence of urinary tract infection was 26.6% [95% CI: 21.5% -31.2%]. The current finding is in line with the study done in Hawassa (23.1%) [35] and Uganda (22.0%) [36]. However, this finding was relatively higher than studies done in Jimma (13.7%) [37], Gojjam Ethiopia (20.6%) [38], Hawassa (4.9%) [39], Addis Ababa, Ethiopia (18.6%) and 11.3% [16, 40] and Harar, Ethiopia (18%) [41]. However, it is lower than studies conducted in Nigeria (32.5%) [42], South Africa (48.7%) [43], Nepal (54.76%) India (100%), and (49.15%) [44, 45]. This variation might be due to differences in sample size, the degree of the immune status of the study participants, geographical variation, ART status, and socioeconomic conditions.

**Table 3. Antimicrobial susceptibility pattern of Gram-negative bacterial isolates from community-acquired UTI among HIV-positive patients at Dessie Comprehensive Specialized Hospital, Dessie, Ethiopia.**

| Etiologic agents | Pattern | Antimicrobial agents N (%) | | | | | | | | | | |
|---|---|---|---|---|---|---|---|---|---|---|---|---|
| | | AMP | AMC | AK | CRO | CIP | GN | CAZ | TE | CTX | SXT | F |
| *E. coli (16)* | S | 4(25) | 3(18.7) | 9(56.2 | 9(56.2 | 8(50) | 10(62.5) | 7(43.8) | 2(12.5) | 8(50) | 2(12.5) | 6(37.5) |
| | R | 12(75) | 13(81.3) | 7(43.8) | 7(43.8) | 8(50) | 6(37.5) | 9(56.2) | 14(87.5) | 8(50) | 14(87.5) | 10(62.5 |
| *K. pneumoniae (11)* | S | 3(27.3) | 4(36.4) | 5(45.5) | 6(54.5) | 5(45.5) | 6(54.5) | 4(36.4) | 1(9.1) | 6(54.5) | 1(9.1) | 4(36.4) |
| | R | 8(72.7 | 7(63.6) | 6(54.5) | 5(45.5) | 6(54.5) | 5(45.5) | 7(63.6) | 10(90.9) | 5(45.5) | 10(90.9) | 7(63.6) |
| *P. mirabilis (5)* | S | 1(20) | 2(40) | 3(60) | 3(60) | 3(60) | 1(20) | 2(40) | 1(20) | 3(60) | 1(20) | 2(40) |
| | R | 4(80) | 3(60) | 2(40) | 2(40) | 2(40) | 4(80) | 3(60) | 4(80) | 2(40) | 4(80) | 3(60) |
| *P. aeruginosa (9)* | S | 3(33.3) | 2(22.2) | 5(55.6) | 5(55.6) | 5(55.6) | 6(66.7) | 4(44.4) | 1(11.1) | 4(44.4) | 2(22.2) | 3(33.3) |
| | R | 6(66.7) | 7(87.8) | 4(44.4) | 4(44.4) | 4(44.4) | 3(33.3) | 5(55.6) | 8(88.9) | 5(55.6) | 7(87.8) | 6(66.7) |
| *Enterobacter cloacae. (4)* | S | 1(25) | 1(25) | 2(50) | 2(50) | 2(50) | 3(75) | 1(25) | 1(25) | 2(50) | 1(25) | 2(50) |
| | R | 3(75) | 3(75) | 2(50) | 2(50) | 2(50) | 1(25) | 3(75) | 3(75) | 2(50) | 3(75) | 2(500) |
| *Providencia spp (6)* | S | 2(33.3) | 2(33.3) | 3(50) | 2(33.3) | 4(66.7) | 4(66.7) | 2(33.3) | 1(16.7) | 3(50) | 1(16.7) | 3(50) |
| | R | 4(66.7) | 4(66.7) | 3(50) | 4(66.7) | 2(33.3) | 2(33.30) | 4(66.7) | 5(83.3) | 3(50) | 5(83.3) | 3(50) |
| Total (51) | S | 14(27.5) | 14(27.5) | 25(49) | 27(52.9) | 27(52.9) | 30(58.8) | 20(39.2) | 7(13.7) | 26(51) | 8(15.7) | 20(39.2) |
| | R | 37(72.5) | 37(72.5) | 26(51) | 24(47.1) | 24(47.1) | 21(41.2) | 31(60.8) | 44(86.3) | 25(49) | 43(84.3) | 31(60.80) |

S: sensitive; R: resistant; AMP: ampicillin; AMC: Amoxicillin-Clavulanic acid; CTX: cefotaxime; CAZ: Ceftazidime; F: Nitrofurantoin; CRO: ceftriaxone; CIP: ciprofloxacin; GN: gentamicin; AK: Amikacin; TE: tetracycline; SXT: Trimethoprim-sulphamethoxazole.

In the current study, urinary tract infection with fungal was 4.9% (17/346). Of the 17 fungal isolates, 47.1% (8/17) were represented by *C. tropicalis*. The study reported on *C. albicans*, *C. glabrata*, and *C. tropicalis* supported the current study with the report of fungal species [46]. Studies showed that within hospital settings at least 11.3% of community-acquired UTIs are caused by *candida species* [47]. In a study conducted in Addis Ababa, Candidauria represented 7% of UTIs in which *C. albicans* was the leading cause and 81.1% of candida species were isolated [22], and a study was done in two microorganisms Candida spp. *(Candida glabrata* and *Candida krusei*) [48].

From the bacterial isolates, 21.33% (16/75) were *E. coli*, the most frequent bacterium, followed by *K. pneumoniae*, *enterococcus species*, and *P. aeruginosa*, accounting for 14.67% (11/

**Table 4. Antimicrobial susceptibility pattern of Gram-positive bacterial isolates from community-acquired UTI among HIV-positive patients attending Dessie Comprehensive Specialized Hospital, Dessie, Ethiopia.**

| Bacterial isolates | | Antimicrobial Agents N (%) | | | | | | | | | |
|---|---|---|---|---|---|---|---|---|---|---|---|
| | | E | P | C | CTX | F | CL | CIP | SXT | FOX | TE |
| | | N (%) | N (%) | N (%) | N (%) | | N (%) | N (%) | N (%) | N (%) | N (%) |
| S. *aureus* (6) | S | 4(66.7) | 1(16.7) | 3(50) | 3(50) | 2(33.3) | 4(66.7) | 4(66.7) | 1(16.7) | 3(50) | 1(16.7) |
| | R | 2(33.3) | 5(83.3) | 3(50) | 3(50) | 4(66.7) | 2(33.3) | 2(33.3) | 5(83.3) | 3(50) | 5(83.3) |
| CoNS (8) | S | 3(37.5) | 2(25) | 2(25) | 3(37.5) | 5(62.5) | 5(62.5) | 2(25) | 1(12.5) | 3(37.5) | 1(12.5) |
| | R | 5(62.5) | 6(75) | 6(75) | 5(62.5) | 3(37.5) | 3(37.5) | 6(75) | 7(87.5) | 5(62.5) | 7(87.5) |
| *Enterococcus species (10)* | S | 4(40) | 3(30) | 5(50) | 6(60) | 3(30) | 7(70) | 6(60) | 2(20) | 4(40) | 1(10) |
| | R | 6(60) | 7(70) | 5(50) | 4(40) | 7(70) | 3(30) | 4(40) | 8(80) | 6(60) | 9(90) |
| Total (24) | S | 11(45.83) | 6(25) | 10(41.7) | 12(50) | 10(41.7) | 16(66.7) | 12(50) | 4(16.7) | 10(41.7) | 3(12.5) |
| | R | 13(54.17) | 18(75) | 14(58.3) | 12(50) | 14(58.3) | 8(33.3) | 12(50) | 20(83.3) | 14(58.3) | 21(87.5) |

**Key: CL** = Clindamycin, **E** = Erythromycin, **C** = Chloramphenicol, **CIP** = Ciprofloxacin, **TE** = Tetracycline, **SXT** = Trimethoprim-Sulfamethoxazole, **CTX** = cefotaxime, **FOX** = Cefoxitin, **F:** nitrofurantoin, **P** = penicillin, **R** = Resistant, **FOX** = Cefoxitin **S** = Sensitive, **NT** = Not tested

**Table 5. Multi-drug resistance patterns of uropathogenic bacterial isolates among UTI symptomatic HIV-positive patients attending Dessie Comprehensive Specialized Hospital, Dessie, Ethiopia (February 1, 2021- March 30, 2021).**

| Isolated organisms | Total | MDR to antimicrobials | | | | | | |
|---|---|---|---|---|---|---|---|---|
| | | R0 N (%) | R1 N (%) | R2 N (%) | R3 N (%) | R4 N (%) | ≥R5 N (%) | MDR N (%) |
| *E. coli* | 16 | 1(6.3) | 1(6.3) | 1(6.3) | 4(25) | 7(43.7) | 2(12.5) | 13(81.3) |
| *K. pneumoniae* | 11 | 1(9.1) | 1(9.1) | - | 4(36.4) | 2(18.2) | 3(27.3) | 9(81.8) |
| *S. aureus* | 6 | 1(16.7) | - | - | 3(50) | 2(33.3) | - | 5(83.3) |
| *P. mirabilis* | 5 | 0 | 1(20) | - | 2(40) | 1(20) | 1(20) | 4(80) |
| *CoNs* | 8 | 1(12.5) | - | - | 4(50) | 2(25) | 1(12.5) | 7(87.5) |
| *P. aeruginosa* | 9 | 0 | 1(11.1) | 1(11.1) | 3(33.3) | 2(22.2) | 2(22.2) | 7(77.8) |
| *Enterobacter Cloacae* | 4 | 0 | 1(25) | - | 1(25) | 2(50) | - | 3(75) |
| *Enterococcus species* | 10 | 1(10) | 1(10) | - | 4(40) | 3(30) | 1(10) | 8(80) |
| *Providencia species* | 6 | 0 | 1(16.7) | - | 2(33.3) | 2(33.3) | 1(16.7) | 5(83.3) |
| Total | 75 | 5(6.7) | 7(9.3) | 2(2.7) | 27(36) | 23(30.7) | 11(14.7) | 61(81.3) |

R0 = No antibiotic resistance category, R1 = Resistance to one antibiotic category, R2 = Resistance to two antibiotics category, R3 = Resistance to three antibiotics category, R4 = Resistance to four antibiotics category, R5 = Resistance to five antibiotics category

75), 13.33% (10/75), and 12% (9/75), respectively. Similarly, *E. coli* was predominant in studies in different parts of Ethiopia and across different countries like, Bahirdar (38.1%) [41], Gondar, (56.1%) [49], and Jimma, (54.3%) [37] from Ethiopia, tertiary care hospital, India (41.7%) [16], Saudi Arabia [50] and Ethiopia [39]. However, it was inconsistent with the findings reported in Ebony State, Nigeria [51] and India [44], they reported that *S. aureus* (45.3%) and *P. aeruginosa* (41.9%), and in Turkey the most common causative agent was *E. coli* (66.6%) followed by *K. pneumoniae* (16.6%) and others such as *Enterobacter spp.* (7.7%) [48] were the commonest urinary tract pathogens. This discrepancy might be due to differences in the test facility, strain variety, geographical variation, and study group variations. The presence of a unique structure that aids bacteria attachment to uroepithelial cells, allowing for multiplication and tissue invasion, could explain the predominance of *E. coli* [52, 53].

Even though no statistically significant association was observed in this study, we found that the bacteriuria was higher among females (27.8%) than males (25.3%). This finding is in agreement with various studies conducted in Ethiopia [40, 41, 49], South Africa [43], and Nigeria [51]. This could be explained by physiological and anatomical differences between males and females, such as males' shorter urethra, a decrease in normal vaginal flora, and females' less acidic vaginal surface pH [40].

In the current study, about 52.9%, 52.9%, and 58.8% of the total isolated Gram-negative Uro-pathogens were susceptible to ceftriaxone, ciprofloxacin, and gentamicin, respectively, whereas, 51%, 72.5%, 72.3%, 84.3%, 60.8%, and 60.8% of the isolates were resistant to amikacin, ampicillin, amoxicillin-Clavulanic acid, and trimethoprim-sulphamethoxazole. The other study on urinary tract bacterial and fungal isolates supported these current findings with frequency percentage sulphamethoxazole, nitrofurantoin, and ceftazidime, and most of the urinary bacterial isolates were highly sensitive to ceftazidime (95%), and ciprofloxacin (88% [54].

In this study, 87.5%, 87.5%, 81.3%, 75%, 62.5%, and 56% of *E. coli* strains were resistant to tetracycline, trimethoprim-sulphamethoxazole, amoxicillin-Clavulanic acid, ampicillin, nitrofurantoin, and ceftazidime, respectively. Similarly, 80% of ampicillin, 90.9% of trimethoprim-sulphamethoxazole, 90.9% of tetracycline, 63.6% of nitrofurantoin, and 63.6% of ceftazidime, 54.5% was resistance to *K. pneumoniae*. There were also similar findings from studies conducted in Addis Ababa which indicated *K. pneumoniae* had the highest level of resistance against trimethoprim-sulphamethoxazole (86.4%) cefotaxime (86.4%), cefepime (85.4%),

**Table 6. Bivariate and multivariable analysis for variables associated with CA-UTI among HIV patients attending Dessie Comprehensive Specialized Hospital, Dessie, Ethiopia (February 1, 2021- march 30, 2021).**

| Characteristics | | Urine Culture Result | | COR (95% CI) | p-value | AOR (95% CI) | P-value |
|---|---|---|---|---|---|---|---|
| | | Positive n (%) | Negative n (%) | | | | |
| Age in years | 18–24 | 10(18.9) | 43(81.1) | 1 | 1 | | |
| | 25–34 | 21(24.7) | 64(75.3) | 0.65(0.25–5.295) | 0.984 | | |
| | 35–44 | 30(25.9) | 86(74.1) | 0.985(0.154–14.038) | 0.838 | | |
| | >44 | 31(33.7) | 61(66.3) | 1.56 (1.11–19.035) | 0.012 | 1.56(1.11–9.35) | 0.001 |
| Sex | Male | 42(25.3) | 124(74.7) | 1 | 1 | 1 | 1 |
| | Female | 50(27.8) | 130(72.2) | 1.79(1.1–3.655) | 0.185 | 3.29(1.26–13.41) | 0.002 |
| Residence | Urban | 62(28.8) | 153(71.2) | 1 | 1 | | |
| | Rural | 30(22.9) | 101(77.1) | 1.4(0.427–3.026) | 0.89 | | |
| Occupation | House Wife | 15(25.9) | 43(74.1) | 1 | 1 | | |
| | Civil Servant | 10(22.2) | 35(77.8) | 0.95(0.41–5.34) | 0.54 | | |
| | Self-Employee | 9(18) | 41(82) | 1.25(0.36–13.54) | 0.36 | | |
| | Driver | 10(20.8) | 38(79.2) | 0.71(0.15–16.12) | 0.65 | | |
| | Daily Labor | 23(25.6) | 67(74.4) | 3.842(1.37–17.91) | 0.17 | 4.54(1.41–21.71) | 0.003 |
| | Farmer | 25(45.5) | 30(55.5) | 4.842(1.17–17.91) | 0.18 | 6.54(1.21–23.51) | 0.004 |
| Educational status | Unable to read and write | 42(36.5) | 73(63.5) | 3.85(1.76–12.61) | 0.114 | 4.58(1.21–13.85) | 0.001 |
| | Primary | 20(23.3) | 66(76.7) | 0.98 (0.928–12.398) | 0.265 | | |
| | Secondary | 22(24.2) | 69(75.8) | 1.95(0.43–12.22) | 0.655 | | |
| | College/University | 8(14.8) | 46(85.2) | 1 | 1 | 1 | 1 |
| Current marital status | Married | 45(37.2) | 76(62.8) | 1.8([0.61–16.86) | 0.43 | | |
| | Unmarried | 18(22.5) | 62(77.5) | 1.369(0.325–5.766) | 0.668 | | |
| | Divorced | 13(13.7) | 82(86.3) | 1.05(0.153–7.270) | 0.956 | | |
| | Widowed | 16(32) | 34(68) | 1 | 1 | | |
| Current CD4+ count (cells/mm$^3$) | ≤200 | 24(32.9) | 49(67.1) | 5.29(1.39–18.51) | **0.018** | 7.44(1.64–27.21) | **0.001** |
| | 201–350 | 35(33.3) | 70(66.7) | 4.18(1.55–17.10) | **0.024** | 4.55(1.36–18.28) | **0.009** |
| | 351–500 | 20(23.8) | 64(762) | 0.98(0.57–12.56) | 0.705 | 0.99(0.89–12.24) | 0.861 |
| | ≥501 | 13(15.3) | 72(84.7) | 1 | 1 | 1 | 1 |
| History of hospitalization | Yes | 35(30.4) | 115(69.6) | 1.98(1.78–13.28) | 0.181 | 6.27(1.25–15.17) | 0.002 |
| | No | 57(29.1) | 139(70.9) | 1 | 1 | 1 | 1 |
| Use of cotrimoxazole as prophylaxis | Yes | 40(26) | 114(74) | 1 | 1 | | |
| | No | 54(29.1) | 138(70.9) | 0.953(0.455–1.995) | 0.898 | | |
| Chronic diabetes | Yes | 32((25.6) | 89(74.4) | 3.78(1.22–15.42) | 0.016 | 4.80(1.85–16.31) | 0.009 |
| | No | 60(24.6) | 184(75.4) | 1 | 1 | 1 | 1 |
| History of catheterization | Yes | 42(30) | 120(70) | 1.794(1.53–16.03) | 0.048 | 4.65(1.55–26.31) | 0.0012 |
| | No | 50(24.3) | 136(75.7) | 1 | 1 | | |
| Dysuria | Yes | 58(27.6) | 152(72.4) | 0.753(0.368–1.979) | 0.711 | | |
| | No | 34(25.4) | 100(74.6) | 1 | 1 | | |
| Frequency of urination | Yes | 55(27.5) | 155(72.5) | 0.629(0.395–1.741) | 0.621 | | |
| | No | 37(25.3) | 109(74.7) | 1 | 1 | | |
| Urgency | Yes | 64(39) | 100(61) | 1.427(1.89–22.55) | 0.039 | 6.51(1.75–36.31) | 0.0025 |
| | No | 28(15.4) | 154(84.6) | 1 | 1 | | |
| Flank pain | Yes | 61(37) | 104(63) | 1.193(0.548–2.599) | 0.657 | | |
| | No | 31(17.1) | 157(82.9) | 1 | 1 | | |

ceftazidime (85.4%), amoxicillin-clavulanic acid (85.4%), gentamicin (70.0%), and ciprofloxacin (50.5% [55], In Iran: trimethoprim-sulphamethoxazole (91.4%), ceftazidime (91.4%), and gentamicin (82.8%) [56]; in Sierra Leone: ciprofloxacin (73.4%) and gentamicin (60%) [57], in Equatorial Guinea: trimethoprim-sulphamethoxazole (100%), amoxicillin-clavulanic acid (96.6%), gentamicin (86.2%) and ciprofloxacin (87.5%) [58]. The occurrence of high antibiotic resistance might be due to misuse and overuse of antibiotics, and poor infection control measures [59]. This was comparable with the study done in Gondar, Ethiopia [49] and Harar, Ethiopia [41], *P. aeruginosa* was 100% resistant to ampicillin, cefotaxime, and gentamicin and trimethoprim-sulfamethoxazole and *P.mirabilis* 100% resistant to ampicillin. This is in agreement with a study done in Ethiopia [40].

Similarly, about 87.5%, 83.3%, 75%, 58.3%, and 58.3% of Gram-positive isolates were resistant to tetracycline, trimethoprim-sulfamethoxazole, penicillin, chloramphenicol, and nitrofurantoin, respectively. The dominant isolated Gram-positive *Enterococcus species* showed more resistance to 90% of tetracycline, 80% of trimethoprim-sulphamethoxazole, and 70% of nitrofurantoin. Likewise, *coagulase-negative staphylococci* were resistant to 87.5% of tetracycline and trimethoprim-sulphamethoxazole, and 75% of penicillin and chloramphenicol, respectively. This is in line with a study done in Harar, Ethiopia, where 57.1% of *S. aureus* were resistant to gentamicin, tetracycline, and cefoxitin (57.1%) [41]. We also found that 85% of *Coagulase-negative staphylococcus spp*. were resistant to trimethoprim-sulphamethoxazole (85%) and over 60% were resistant to penicillin, gentamicin, and tetracycline.

Multidrug resistance has serious implications for the health outcomes of HIV-infected patients [60]. It is quite alarming to note that almost 61(81.3%) of currently isolated bacteria were found to be resistant to three or more antimicrobials. This was higher compared to previous findings reported in Harar, Ethiopia (46%) [41], Dessie (74.6%) [61], Gondar (68%) [62], and Nepal (64.04%) [63] and India (28%) [64]. But it was lower than a report from Gondar, Ethiopia (95%) and 87.4% [12, 49], Addis Ababa (100%) [40], Bahirdar (93.1%) [65], Nepal (96.84%) [63]. These differences might be due to the irrational drug utilization habit of the communities or to the distribution of those sensitive and resistant strains of bacteria.

There was an intra-species difference in MDR level. The present study showed that the level of MDR in *E. coli* 13 (81.3%), *K. pneumoniae* 9(81.8%), *Enterococcus species* 8(80%), *P. aeruginosa* 7 (77.8%), and CoNs 7(87.5%). This was comparable with a study conducted in Addis Ababa *K. pneumoniae* (83.5%) and *E. coli* (66.2%) [55], in Equatorial Guinea *E. coli* (74.4%) [58]; in Sierra Leone *K. pneumoniae* (73.3%) and *E. coli* (61.5%) [57]; in Gondar, *K. pneumoniae* was (95.6%) and *E. coli* was (92.9%) [12]; in Khartoum-Sudan *E. coli* (92.2%) [66]; and in Equatorial Guinea in *K. pneumoniae* (91.7%) [58]. The MDR level among E. coli (50.2%) in Dessie is lower than in our study [61]. The difference in MDR levels between *K. pneumoniae* and *E. coli* in our study might be most K. pneumoniae were isolated from blood specimens that were collected from inpatients in the hospitals.

In this study, there was no statistically significant association between significant bacteriuria and patients' residence, marital status, dysuria, frequency of urination, flank pain, and use of co-trimoxazole. However, age, gender, occupation, educational status, current CD4+ count (cells/mm3), previous hospitalization and diabetes status, history of catheterizations, and urinary tract urgency had a statistically significant association with significant bacteriuria and Candidauria. Similar findings were reported from Jimma, Addis Ababa, and Harar [37, 40, 41]. Furthermore, age groups >44year [AOR = 1.56, 95%CI: 1.11–9.35, P = 0.001], being female [AOR = 3.29, 95%CI: 1.26–13.41, P = 0.002], being daily labor [AOR = 4.54, 95%CI: 1.41–21.71, P = 0.003], being farmer [AOR = 6.54, 95%CI: 1.21–23.51, P = 0.004], participants unable to read and write [AOR = 4.58, 95%CI: 1.21–13.58, P = 0.001], HIV patients with CD4 count of ≤ 200 cells/mm3 (AOR: 7.44; 95%CI: 1.64–27.21, P = 0.001) and CD4 count of 201–

350 cells/mm3 (AOR: 4.55; 95%CI: 1.36–18.28, P = 0.009), patients who had chronic diabetics (AOR: 4.80; 95%CI: 1.85–16.31, p = 0.009), patients having a history of hospitalization (AOR: 6.27; 95%CI: 1.25–15.17, p = 0.002) and patients who had the urgency of urination (AOR: 6.51; 95%CI: 1.75–36.31, p = 0.0025). Studies conducted in Nepal support our findings with a significant risk factor for community-acquired urinary tract infections in patients with indwelling urinary catheters (38.3%), diabetes mellitus (36%), malignancies (5.2%), and urinary incontinence (11.7%). Urinary catheters are a proven source of infection, especially when they are kept for longer periods [47]. Similarly, research finding was supported by studies conducted elsewhere [22, 26, 27, 29]. The explanation for the inverse relationship between UTI and a CD4+ count is unknown; it is probably due to the impaired immunity at a declining CD4 + count that makes it easier for bacterial pathogens to adhere to the urinary epithelium.

## Limitations of the study

The study was a single hospital-based study and might not represent all HIV-infected patients and did not include a control group due to budget constraints. We did not attempt to identify other causative agents (anaerobic bacteria, and fungus) that would have made a significant contribution to a true prevalence of UTI in HIV-positive patients.

## Conclusions and recommendations

The symptomatic community-acquired UTI caused by bacterial pathogens and fungi is high, as compared to different national and local data. Of the isolated pathogens, bacteria were high as compared to fungal pathogens. Among bacterial pathogens, *E. coli* is the most frequent bacterium, followed by *K. pneumoniae*, *enterococcus species*, and *P. aeruginosa*, accounting for, respectively. Of the isolated bacteria, 61(81.3%) were resistant to three and above classes of antibiotics (drug classes), and *E. coli*, *K. pneumoniae*, *Enterococcus species*, *P. aeruginosa*, and *CoNs* showed high levels of drug resistance. Most Gram-negative and Gram-positive isolates were resistant to tetracycline, trimethoprim-sulfamethoxazole, penicillin, chloramphenicol, and nitrofurantoin, respectively. However, uropathogenic were susceptible to amikacin, it was more effective against these pathogens. Patients who had low CD4 counts, previous hospitalization, diabetes status, and having history of catheterizations than their counterparts were more at risk of developing bacterial urinary tract infections. Management of UTI among symptomatic HIV patients should be supported by laboratory results of urine culture. Regular monitoring of antimicrobial resistance patterns should be undertaken by healthcare providers. Individuals at risk for bacterial-associated UTI should have good adherence to ART and their CD4 levels should be monitored regularly. Health information dissemination should be given about UTIs and the habit of drug use for HIV-positive patients.

## Acknowledgments

The authors would like to acknowledge the Department of Medical Laboratory Science, College of Medicine and Health Sciences, Wollo University for providing the laboratory setup and facilities to conduct the experiments. Dessie Comprehensive Specialized Hospital and all study participants are gratefully acknowledged for their kind cooperation during data collection.

## Author Contributions

**Conceptualization:** Mihret Tilahun, Mihreteab Alebachew, Alemu Gedefie, Melaku Ashagrie Belete, Abdurahaman Seid, Daniel Gebretsadik, Ermiyas Alemayehu, Wondmagegn Demsiss, Agumas Shibabaw, Habtamu Mekonnen, Tewodros Dessie.

**Data curation:** Mihret Tilahun, Mihreteab Alebachew, Alemu Gedefie, Melkam Tesfaye, Abdurahaman Seid, Agumas Shibabaw, Tewodros Dessie.

**Formal analysis:** Mihret Tilahun, Melkam Tesfaye, Ermiyas Alemayehu, Wondmagegn Demsiss, Habtamu Mekonnen.

**Funding acquisition:** Mihret Tilahun, Daniel Gebretsadik, Agumas Shibabaw.

**Investigation:** Mihret Tilahun, Alemu Gedefie, Endris Ebrahim, Melkam Tesfaye, Melaku Ashagrie Belete, Daniel Gebretsadik, Ermiyas Alemayehu, Bekele Sharew, Agumas Shibabaw, Tewodros Dessie.

**Methodology:** Mihret Tilahun, Mesfin Fiseha, Mihreteab Alebachew, Endris Ebrahim, Melaku Ashagrie Belete, Abdurahaman Seid, Daniel Gebretsadik, Ermiyas Alemayehu, Wondmagegn Demsiss, Bekele Sharew, Habtamu Mekonnen, Tewodros Dessie.

**Project administration:** Mihret Tilahun.

**Resources:** Mihret Tilahun.

**Software:** Mihret Tilahun.

**Supervision:** Mihret Tilahun, Habtamu Mekonnen.

**Validation:** Mihret Tilahun, Mesfin Fiseha, Abdurahaman Seid.

**Visualization:** Mihret Tilahun, Mesfin Fiseha, Melaku Ashagrie Belete, Abdurahaman Seid.

**Writing – original draft:** Mihret Tilahun, Mesfin Fiseha, Endris Ebrahim, Melkam Tesfaye, Melaku Ashagrie Belete, Abdurahaman Seid, Daniel Gebretsadik, Ermiyas Alemayehu, Wondmagegn Demsiss, Bekele Sharew, Agumas Shibabaw, Tewodros Dessie.

**Writing – review & editing:** Mihret Tilahun, Mesfin Fiseha, Daniel Gebretsadik, Wondmagegn Demsiss, Bekele Sharew, Agumas Shibabaw, Tewodros Dessie.

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
