## [Decision Letter · Decision Letter 0]

11 Apr 2023

PONE-D-22-31717

Bacterial and fungal profile, multidrug resistance profile of bacterial pathogens and associated risk factors of community-acquired urinary tract infection among HIV positive patients attending Antiretroviral Therapy Clinics of Northeast Ethiopia.

PLOS ONE

Dear Dr. Tilahun,

Thank you for submitting your manuscript to PLOS ONE. After careful consideration, we feel that it has merit but does not fully meet PLOS ONE’s publication criteria as it currently stands. Therefore, we invite you to submit a revised version of the manuscript that addresses the points raised during the review process.

We look forward to receiving your revised manuscript.

Kind regards,

Eric Sampane-Donkor

Academic Editor

PLOS ONE

- http://etd.aau.edu.et/bitstream/handle/123456789/6042/Yemisrach%20Getu.pdf?

In your revision ensure you cite all your sources (including your own works), and quote or rephrase any duplicated text outside the methods section. Further consideration is dependent on these concerns being addressed.

A clean copy of the edited manuscript (uploaded as the new *manuscript* file).

“All the authors are under developing countries and no extrena funding was obtained”

Additional Editor Comments:

The reviewers have raised several deficiencies about the manuscript, which require attention. Kindly address these deficiencies and resubmit the manuscript.

Reviewers' comments:

Reviewer's Responses to Questions

**Comments to the Author**

1. Is the manuscript technically sound, and do the data support the conclusions?

Reviewer #1: Partly

Reviewer #2: Yes

2. Has the statistical analysis been performed appropriately and rigorously? 

Reviewer #1: Yes

Reviewer #2: Yes

3. Have the authors made all data underlying the findings in their manuscript fully available?

Reviewer #1: Yes

Reviewer #2: Yes

4. Is the manuscript presented in an intelligible fashion and written in standard English?

Reviewer #1: No

Reviewer #2: No

5. Review Comments to the Author

Reviewer #1: Check the sticky note on your PDF file. Try to address all the comments. I have tried to your entire manuscript it lacks basic thing. if modified and edited with expert in the field it match improved. But I saw some errors that makes me to suspicious on your data. Rewrite the introduction part and discussion part seriously.

Reviewer #2: In this article, the authors have reported relevant data on the etiology and susceptibility profile of Community-Acquired UTI among High risk group-HIV patients in Ethiopia. Key findings precipitated by this study are:

1. The prevalence of Community-Acquired UTI among HIV patients in the study was high (29%)

2. Bacteria were the most implicated microorganisms causing UTI among the study population with E. coli being the most isolated bacteria

3. The prevalence of MDR bacteria was quite high in this study (>80%)

4. The risk of Community-Acquired UTI among HIV patients was found to be associated with low CD4 counts, diabetes, history of hospitalization and catheterization.

The methods and design described by the authors appropriately address the research aim.

The conclusion, the limitations and recommendations are clear and supported by the results reported.

Although addressing an important issue of global health concern, the study reported however, lacks a novel feature. This is because, there exist, quite a number of studies addressing the etiology, and risk factors of UTI among HIV patients in Ethiopia. That notwithstanding, the addition of MDR assessment in this study is commendable.

Of note, this manuscript has a number of issues to be addressed:

1. The title of this manuscript is quite lengthy. Authors should consider shortening the title to a concise one. Please consider, “Multidrug resistance and associated risk factors of bacterial- and fungal UTI among HIV patients attending Antiretroviral Therapy Clinics of Northeast Ethiopia”.

2. The manuscript has a significant number of omissions, typographical and grammatical errors, and incoherent sentences. Authors will need to carefully go through the manuscript for corrections or employ the services of a script/copy editor.

3. All genus and specie names of microorganisms should be italicized. This is not consistent in the manuscript.

4. All abbreviations should be written in full in the places where they are first used. This is inconsistent in the manuscript.

INTRODUCTION

1. In the introduction, the authors highlight that ‘no related study was done in the study area in particular’ and hence the need for this study. It would be appropriate to make reference to findings of similar studies done in other sites of similar settings in;

A. Ethiopia,

B. Resource-limited areas of the East African Subregion.

2. Although the authors have given some justification for their research, a definite Statement of Problem(s) addressed by their research is not clearly stated in the Introduction.

3. Additionally, the authors should enumerate the specific objectives of their study in the Introduction

4. The literature review provided for MDR in the Introduction is scanty.

5. The Introduction should include at least, a sentence or a paragraph on each of the following;

a. Definition, Symptoms and Types of UTI

b. Epidemiology of UTI in Ethiopia, East Africa, and Sub-saharan Africa

c. Epidemiology of HIV in Sub-saharan Africa and Ethiopia

d. Prevalence and Risk of UTI among HIV patients

e. Factors associated with urinary tract infections among HIV Patients (please see Skrzat-Klapaczyńska A, Matłosz B, Bednarska A, et al. Factors associated with urinary tract infections among HIV-1 infected patients. PLoS One. 2018;13(1):e0190564. Published 2018 Jan 11. doi:10.1371/journal.pone.0190564)

Results and Disussion:

1. Can the Authors explain why they could not establish any association between bacteriuria and symptoms of UTI in their study?

2. Please provide a reference to this statement in the Discussion (pg 15, paragraph 1) ; Multiple studies showed that within hospital settings at least 10–15% of hospital-acquired UTIs are caused by Candida species and Candidauria is especially common in intensive care units

Minor Considerations:

The section in the Study site description can be shortened

Titles of Tables can be shortened

Authors should consider, ‘ Etiology of UTI among HIV patients attending Antiretroviral Therapy Clinics of Northeast Ethiopia’ as the title for Figure 1.

6. PLOS authors have the option to publish the peer review history of their article (what does this mean?). If published, this will include your full peer review and any attached files.

Reviewer #1: **Yes: **Tsegaye Alemayehu (assistant Professor)

Reviewer #2: **Yes: **Samuel Darkwah

---

## [Author Response · Author response to Decision Letter 0]

21 Apr 2023

Author’s Response to Editor and Reviewer comments

Manuscript ID number:

PONE-D- 22-31717

Title of paper:

Bacterial and fungal profile, multidrug resistance profile of bacterial pathogens and associated risk factors of community-acquired urinary tract infection among HIV positive patients attending Antiretroviral Therapy Clinics of Northeast Ethiopia

 General

We appreciate all of the constructive feedback on our manuscript from the reviewers, which has helped us improve the paper's clarity and scientific plausibility. We thank the PLOS ONE scholarly editors for their insightful comments and for allowing us to update the text. All of the concerns raised by the reviewers and editors have been addressed, and the text has been updated as a result. Furthermore, both the updated article and the response letter address all of the editorial revisions that are required. The changes are shown with track changes in the file labelled "Revised Manuscript with Track Modifications." The following is a detailed rebuttal.

Editor Comments:

Author’s Response: 

 Thanks for the substantial editorial comment. The revised manuscript is now updated, highlighted and meets PLOS ONE's style requirements, including those for file naming.

2. We noticed you have some minor occurrence of overlapping text with the following previous publication(s), which needs to be addressed: - http://etd.aau.edu.et/bitstream/handle/123456789/6042/Yemisrach%20Getu.pdf? In your revision ensure you cite all your sources (including your own works), and quote or rephrase any duplicated text outside the methods section. Further consideration is dependent on these concerns being addressed

Author’s Response: 

 Thank you for the interesting comment. Minor over lapping’s are corrected 

 ” Author’s Response: 

 Thank you and corrections are made as per the comment; all the authors are senior expert and published many papers and one of the authors Melaku Ashagire Belete is the editors of plose one. So no need of language editors.

4. Thank you for stating the following financial disclosure: “All the authors are under developing countries and no extrena funding was obtained” At this time, please address the following querie.

Author’s Response: 

 Thank you for the interesting comment. financial disclosure corrected as No external funds were obtained only internal support was obtained from Wollo university. The funders had no role in study design, data collection and analysis, decision to publish, or preparation of the manuscript

Reviewer #1 comments

Reviewer #1: Check the sticky note on your PDF file. Try to address all the comments. I have tried to your entire manuscript it lacks basic thing. if modified and edited with expert in the field it match improved. But I saw some errors that makes me to suspicious on your data. Rewrite the introduction part and discussion part seriously.

Author’s Response: 

 Thanks, and we have amended all comments as per request and highlighted and some questions and comments are here below 

The date was collected from patients before 48 hours. The data written 18 hours is clerical error 

1. Write Bacterial and fungal profile as uropathogens, multidrug resistance profile rephrase of bacterial pathogens and associated risk factors as associated factorstof community-acquired urinary tract infection among HIV positive patients attending Antiretroviral Therapy 

Clinics of Northeast Ethiopia

Author’s Response 

We modify the title Uropathogens, multidrug resistance pattern of bacterial and associated factors of community-acquired urinary tract infection among HIV positive patients attending Antiretroviral Therapy Clinics of Northeast Ethiopia.

2. Source of standardized questionaries was where? And why not performed Recent CD4 count? 

Author’s Response: 

 Thanks for the vivid comment. The source of questionaries or we were developed the questionaries by modifications from WHO Suspected Urinary Tract Infection patient questionnaire and from other published articles 

 CD4+ cell counts are usually measured when you are diagnosed with HIV (at baseline), every 3 to 6 months during first 2 years or until your CD4 count increases above 300 cells/mm3. So it is better to take from their medical records those who have less than six months

3. Can you justify do you collect mid steam urine or catheterized patients? What about significance .

Author’s Response: 

 Thank you for the substantial comment. We have collected mid-stream urine and symptomatic patients so the interpretations are Symptomatic ambulatory patient; voided or clean-catch specimen: report if growth is ≥ 10,000 CFU/ml with 1-2 species of potential pathogen. 

4. On the result section all comments are corrected per comments and sugestions but the same isolates are 74 not 75 ?

Author’s Response: 

 Thanks for the valuable Comment and you can calculated and subtract again 92-17 =75 

5. Result and discussion: how do you identify C.tropicalis from C.albican 

Author’s Response: 

 Thank you for the valuable comment. Gram stain, Germ Tube Test, Carbohydrate Fermentation, and Assimilation Tests, as well as chromogenic medium (CHROMagar Candida Medium (Biomerieux, France) according to the manufacturer's instructions .

 Perry and Miller reported that Candida albicans produces an enzyme b -N-acetyl- galactosaminidase and according to Rousselle et al incorporation of chromogenic or fluorogenic hexosaminidase substrates into the growth medium helps in identification of C. albicans isolates directly on primary isolation. HiCrome Candida Differential Agar is a selective and differential medium, which facilitates rapid isolation of yeasts from mixed cultures and allows differentiation of Candida species namely C. albicans, C. krusei, C. tropicalis and C. glabrata on the basis of coloration and colony morphology. On this medium results are obtained within 48 hours and it is useful for the rapid and presumptive identification of common yeasts in Mycology and Clinical Microbiology Laboratory. Peptone special and yeast extract provides nitrogenous, carbonaceous compounds and other essential growth nutrients. Phosphate buffers the medium well. Chloramphenicol suppresses the accompanying bacterial flora. C. albicans appear as light green colored smooth colonies, C. tropicalis appear as blue to metallic blue colored raised colonies. C. glabrata colonies appear as cream to white smooth colonies, while C. krusei appear as purple fuzzy colonies.

6. Rewrite MDR based on guideline of for table construct from journal for submisions. 

Author’s Response: 

 Thanks, and we have amended the conclusion part as we write on the A bacterium that is simultaneously resistant for three or more antimicrobial categories. I understand your questions but to write each bacteria it is very length. 

7. Using references 23 From Hawassa are not ART 

Author’s Response: 

We use Hawassa non-HIV patients as a reference because both HIV and Diabetics are chronic diseases and they lead patients immunocompromised 

Reviewer #1: Check the sticky note on your PDF file. Try to address all the comments. I have tried to your entire manuscript it lacks basic thing. if modified and edited with expert in the field it match improved. But I saw some errors that makes me to suspicious on your data. Rewrite the introduction part and discussion part seriously.

Author’s Response: 

 Thanks, and we have amended all comments as per request and highlighted 

Reviewer #2 comments

1. The title of this manuscript is quite lengthy. Authors should consider shortening the title to a concise one. Please consider, “Multidrug resistance and associated risk factors of bacterial- and fungal UTI among HIV patients attending Antiretroviral Therapy Clinics of Northeast Ethiopia”.

Author’s Response: 

 Comments well taken and the manuscript corrected accordingly, thanks. 

2. The manuscript has a significant number of omissions, typographical and grammatical errors, and incoherent sentences. Authors will need to carefully go through the manuscript for corrections or employ the services of a script/copy editor.

Author’s Response: 

 Comment addressed and highlighted, thanks

3. All genus and specie names of microorganisms should be italicized. This is not consistent in the manuscript.

Author’s Response: 

 Comment addressed and amended, microorganisms should be italicized and highlighted . 

4. All abbreviations should be written in full in the places where they are first used. This is inconsistent in the manuscript.

Author’s Response: 

 Thanks; we really appreciate the comment which helped us to see the gap. Hence, we revised the sentence , included it under and highlighted. 

5. In the introduction, the authors highlight that ‘no related study was done in the study area in particular’ and hence the need for this study. It would be appropriate to make reference to findings of similar studies done in other sites of similar settings in;

A.Ethiopia,

B. Resource-limited areas of the East African Subregion.

Author’s Response: 

 Thanks for critical comments we have to search as we can and we siad that there is limited research in the study area means around dessie comprehensive specialized catchment area. 

6. Additionally, the authors should enumerate the specific objectives of their study in the Introduction 

Author’s Response: 

Comment addressed and amended, thanks. The main objective covers the specific objectives. this study was to determine the bacterial and fungal profiles, multidrug resistance profiles of bacterial pathogens, and associated risk factors for community-acquired urinary tract infection among HIV positive patients attending Antiretroviral Therapy Clinics in Northeast Ethiopia. If we write every spesfic objective it idea redundancy. 

7. The literature review provided for MDR in the Introduction is scanty.

 Author’s Response: 

 Thanks for interesting Comments and we are addressed like Many research studies in the world and Africa, including different parts of Ethiopia, showed that more than 50% of uro-pathogenic bacteria developed multidrug resistance (MDR). About 74% of uro-pathogenic bacteria (6) 70.0% of bacteria from India (7), 79.3% of uro-pathogenic bacterial isolates from Asia Wolayta Sodo Ethiopia (8), in Addis Ababa, 78.4% of Ethiopians (9) and in Gondar, Ethiopia, 91.7% (10) developed MDR thanks 

8. The Introduction should include at least, a sentence or a paragraph on each of the following;

a. Definition, Symptoms and Types of UTI

b. Epidemiology of UTI in Ethiopia, East Africa, and Sub-saharan Africa

c. Epidemiology of HIV in Sub-saharan Africa and Ethiopia

d. Prevalence and Risk of UTI among HIV patients

e. Factors associated with urinary tract infections among HIV Patients (please see Skrzat-Klapaczyńska A, Matłosz B, Bednarska A, et al. Factors associated with urinary tract infections among HIV-1 infected patients. PLoS One. 2018;13(1):e0190564. Published 2018 Jan 11. doi:10.1371/journal.pone.0190564)

 Author’s Response: 

 Thanks for motivating Comments and we are adding the definitions of UTI as follows Urinary tract infection (UTI) is an infection of urinary tract, including the kidneys, ureters and bladder, urethra, and accessory structures that collect and store urine and release it from the body. It occurs when microorganisms, typically bacteria from the digestive tract, cling to the urethral opening and begin to multiply. Globally, the burden of UTI is estimated to be 150 million cases every year (3). Based on previous epidemiological studies from Tanzania and elsewhere in sub-Saharan Africa, the urinary tract infections was estimated at 35–45% (4). In Ethiopia the pooled prevalence of urinary tract infection was 15.97%(5). Urinary tract infections in HIV patients in Ethiopia was 12.8%

 And we have added the recommended reference An other study on Factors associated with urinary tract infections among HIV-1 infected patients also showed that low CD4 count, chronic hepatitis, being female have been linked to occurrence of bacterial UTI (22)

9. Please provide a reference to this statement in the Discussion (pg 15, paragraph 1) ; Multiple studies showed that within hospital settings at least 10–15% of hospital-acquired UTIs are caused by Candida species and Candidauria is especially common in intensive care units

Author’s Response: 

 Thanks for motivating Comments you can observe reference number 47 ,22 and 48 

10. The section in the Study site description can be shortened, Titles of Tables can be shortened

Author’s Response: 

all The titles are modified to Uropathogens, multidrug resistance pattern of bacterial and associated factors of community-acquired urinary tract infection among HIV positive patients attending Antiretroviral Therapy Clinics of Northeast Ethiopia and Section of study are shortened based on the comments 

the table name and figure legends are corrected based on recommendations.

Finally, we thank you for your critical review of the paper.

 Sincerely,

 Mihret Tilahun

---

## [Decision Letter · Decision Letter 1]

28 Jul 2023

PONE-D-22-31717R1Uropathogens, multidrug resistance pattern of bacterial and associated factors of community-acquired urinary tract infection among HIV positive patients attending Antiretroviral Therapy Clinics of Northeast EthiopiaPLOS ONE

Dear Dr. Tadesse,

Thank you for submitting your manuscript to PLOS ONE. After careful consideration, we feel that it has merit but does not fully meet PLOS ONE’s publication criteria as it currently stands. Therefore, we invite you to submit a revised version of the manuscript that addresses the points raised during the review process.

The manuscript need language editor.

the laboratory methods must be standard mainly the drug resistance test.  

We look forward to receiving your revised manuscript.

Kind regards,

Mengistu Hailemariam Zenebe, PhD

Academic Editor

PLOS ONE

Additional Editor Comments:

Dear Author,

This manuscript have an interesting set of findings that I enjoyed reading. However, I thought that the manuscript's weak methodology undermined the findings and conclusions. The reviewers have stated their objections that require significant adjustment. In order to make the study more fascinating and less confirming, answer all the questions that have been posed and do so appropriately.

Reviewers' comments:

Reviewer's Responses to Questions

**Comments to the Author**

1. If the authors have adequately addressed your comments raised in a previous round of review and you feel that this manuscript is now acceptable for publication, you may indicate that here to bypass the “Comments to the Author” section, enter your conflict of interest statement in the “Confidential to Editor” section, and submit your "Accept" recommendation.

Reviewer #2: (No Response)

Reviewer #3: (No Response)

2. Is the manuscript technically sound, and do the data support the conclusions?

Reviewer #2: Yes

Reviewer #3: Partly

3. Has the statistical analysis been performed appropriately and rigorously? 

Reviewer #2: Yes

Reviewer #3: I Don't Know

4. Have the authors made all data underlying the findings in their manuscript fully available?

Reviewer #2: Yes

Reviewer #3: Yes

5. Is the manuscript presented in an intelligible fashion and written in standard English?

Reviewer #2: No

Reviewer #3: No

6. Review Comments to the Author

Reviewer #2: PONE-D-22-31717R1

Title: Uropathogens, multidrug resistance pattern of bacterial and associated factors of community-acquired urinary tract infection among HIV positive patients attending Antiretroviral Therapy Clinics of Northeast Ethiopia

Overview:

The authors have reported relevant data on the etiology and susceptibility profile of Community-Acquired UTI among High risk group-HIV patients in Ethiopia. Key findings precipitated by this study are:

The prevalence of Community-Acquired UTI among HIV patients in the study was high (26.6%)

Bacteria were the most implicated microorganisms causing UTI among the study population with E. coli being the most isolated bacteria

The prevalence of MDR bacteria was quite high in this study (>80%)

The risk of Community-Acquired UTI among HIV patients was found to be associated with low CD4 counts, diabetes, history of hospitalization and catheterization,

The methods and design described by the authors appropriately address the research aim

In conclusion, the limitations and recommendations are clear and supported by the results reported.

The authors have addressed most of the critical issues addressed in the previous review. However, a few things are left to be resolved:

1. The new title of this manuscript is confusing and still lengthy. Authors should consider this title: Please consider, “Uropathogens: multidrug resistance and risk factors of community-acquired UTI among HIV patients attending antiretroviral therapy clinics of Northeast Ethiopia”.

2. The manuscript still has a significant number of omissions, typographical and grammatical errors. Authors will need to carefully go through the manuscript for thorough corrections or better still, employ the services of a script/copy editor. (eg. Another not “An-other”; caused not “causes”; CT-scan not “city-scan”; use of upper-case within sentences; etc…)

3. All genus and specie names of microorganisms should be italicized. This remains pesistent in the manuscript (eg. Fungi species like C. albicans, C. glabrata, and C. tropicalis, in several parts of the manuscript and are not italicized).

4. The authors have not discussed why they could not establish any association between bacteriuria and symptoms of UTI in their study.

Minor Considerations:

5. Titles of Tables can further be shortened.

Reviewer #3: (No Response)

7. PLOS authors have the option to publish the peer review history of their article (what does this mean?). If published, this will include your full peer review and any attached files.

Reviewer #2: **Yes: **Samuel Darkwah

Reviewer #3: **Yes: **Prof. Rambir Singh, Mizoram University, Aizawl, INDIA

---

## [Author Response · Author response to Decision Letter 1]

15 Sep 2023

If there is comments I will egger to revise

---

## [Decision Letter · Decision Letter 2]

8 Oct 2023

PONE-D-22-31717R2Uropathogens: Multidrug resistance and risk factors of community-acquired UTI among HIV patients attending antiretroviral therapy clinics of Northeast EthiopiaPLOS ONE

Dear Dr. Tilahun,

Thank you for submitting your manuscript to PLOS ONE. After careful consideration, we feel that it has merit but does not fully meet PLOS ONE’s publication criteria as it currently stands. Therefore, we invite you to submit a revised version of the manuscript that addresses the points raised during the review process.

We look forward to receiving your revised manuscript.

Kind regards,

Mengistu Hailemariam Zenebe, PhD

Academic Editor

PLOS ONE

**Additional Editor Comments:**

Dear Author, please look at the given comments by the reviewer and respond point by point Indicate which changes you require for acceptance versus which changes you recommendAddress any conflicts between the reviews so that it's clear which advice the authors should followProvide specific feedback from your evaluation of the manuscriptPlease ensure that your decision is justified on PLOS ONE’s publication criteria and not, for example, on novelty or perceived impact.

Reviewers' comments:

Reviewer's Responses to Questions

**Comments to the Author**

1. If the authors have adequately addressed your comments raised in a previous round of review and you feel that this manuscript is now acceptable for publication, you may indicate that here to bypass the “Comments to the Author” section, enter your conflict of interest statement in the “Confidential to Editor” section, and submit your "Accept" recommendation.

Reviewer #4: All comments have been addressed

Reviewer #5: (No Response)

2. Is the manuscript technically sound, and do the data support the conclusions?

Reviewer #4: Yes

Reviewer #5: Partly

3. Has the statistical analysis been performed appropriately and rigorously? 

Reviewer #4: Yes

Reviewer #5: Yes

4. Have the authors made all data underlying the findings in their manuscript fully available?

Reviewer #4: Yes

Reviewer #5: Yes

5. Is the manuscript presented in an intelligible fashion and written in standard English?

Reviewer #4: No

Reviewer #5: No

6. Review Comments to the Author

Reviewer #4: Overall, the paper has been substantially revised from the first revision. Yet, it needs language editions and grammatical corrections. Besides, moderate revisions are recommended before publication (details attached).

Reviewer #5: General comment

The research area is one of an important topic for public health and thank you for addressing that. Here I have some comments and queries to improve the manuscript

it is confusing to identify which one is the recent document, I just saw the first one

Generally the document requires a significant grammatical and typo correction, I recommend an English native or expert to revise the write-up

Please write scientific names correctly

Check for plagiarism, there is a significant copy

Follow the journal guideline strictly, in all aspects

Tables and figures are not of publication quality

Better to modify the title to: Uropathogens, drug resistance and associated factors of community acquired UTI among patients on ART, Northeast Ethiopia

Abstract: Objective: - why only multi-drug resistance? I think it will be better if you just say drug-resistance

In your conclusion it says asymptomatic, correct or include in your objectives

On the third statement it mention about symptomatic UTI: which one was your population

Introduction: Please include this article https://doi.org/10.1371/journal.pone.0264732

There is a significant typo and grammatical error.

Check for plagiarism

Methods: Population: ….. before 18 hr of admission… : it is not clear. If patients come for admission they will be admitted when they visit especially ART patients with known status and followup

Eligibility criteria: it is not clear: is it antibiotics of antibacterial? How about anti-fungal drugs

Mostly ART patients will take drugs for prophylactics

Variables: one of your exclusion criteria is antibacterial use and yet you have included cotri usage as variable

sample-size: I would not agree on the calculation as it does not consider risk factors, and sampling technique is not indicated

Data: CD4 count: was it a recent cunt or old one?

Result

Make sure the tables are in accordance with the journal requirement

Drug use, hospitalization and catheter should be time bounded, When?

Prevalence: in the bacterial isolates the total is not 100% (1.34%) missing, Where there any sample with more than one isolates

What is your cut point to conclude that a drug is effective for uropathogens or not based on your finding? Amikacin and Gentamicin were recommended in the abstract

Discussion

compare your result with a study having similar scope

when you compare studies please consider the confidence interval

Page 14: the justification for result variation is not supported with evidence

limitation of the study

When writing the limitation consider the scope of your study

Conclusion: if symptomatic UTI is supported by Culture result what do you mean by regular AMR followup

7. PLOS authors have the option to publish the peer review history of their article (what does this mean?). If published, this will include your full peer review and any attached files.

Reviewer #4: **Yes: **Hylemariam Mihiretie Mengist

Reviewer #5: No

---

## [Author Response · Author response to Decision Letter 2]

16 Nov 2023

All the comments of Editors and reviewrs including languages are corrected based the recommendations and highlighted

---

## [Decision Letter · Decision Letter 3]

15 Dec 2023

Uro-pathogens: Multidrug resistance and associated factors of community-acquired UTI among HIV patients attending antiretroviral therapy in Dessie Comprehensive Specialized Hospital, Northeast Ethiopia

PONE-D-22-31717R3

We’re pleased to inform you that your manuscript has been judged scientifically suitable for publication and will be formally accepted for publication once it meets all outstanding technical requirements.

Kind regards,

Mengistu Hailemariam Zenebe, PhD

Academic Editor

PLOS ONE

Additional Editor Comments (optional):

Dear Author,

please correct the given comment mainly in siting the reference for sample collection tool.

Best

Reviewers' comments:

Reviewer's Responses to Questions

**Comments to the Author**

1. If the authors have adequately addressed your comments raised in a previous round of review and you feel that this manuscript is now acceptable for publication, you may indicate that here to bypass the “Comments to the Author” section, enter your conflict of interest statement in the “Confidential to Editor” section, and submit your "Accept" recommendation.

Reviewer #4: All comments have been addressed

2. Is the manuscript technically sound, and do the data support the conclusions?

Reviewer #4: Yes

3. Has the statistical analysis been performed appropriately and rigorously? 

Reviewer #4: Yes

4. Have the authors made all data underlying the findings in their manuscript fully available?

Reviewer #4: Yes

5. Is the manuscript presented in an intelligible fashion and written in standard English?

Reviewer #4: Yes

6. Review Comments to the Author

Reviewer #4: Hi there,

Thank you for considering my comments exhaustively. I understand you can't rectify some comments through revisions like the reference used for sample size calculation and data on Methicillin resistance of CoNS. I advise authors to re-edit and rephrase the language for better readability of the paper. Besides, make your tables smart by bordering only at the top and bottom margins. Otherwise ready for acceptance and no need to respond to my views at this time.

7. PLOS authors have the option to publish the peer review history of their article (what does this mean?). If published, this will include your full peer review and any attached files.

Reviewer #4: **Yes: **Hylemariam Mihiretire Mengist

---

## [Editor Report · Acceptance letter]

10 May 2024

PONE-D-22-31717R3 

PLOS ONE

Dear Dr. Tilahun, 

I'm pleased to inform you that your manuscript has been deemed suitable for publication in PLOS ONE. Congratulations! Your manuscript is now being handed over to our production team.

Kind regards, 

on behalf of

Dr. Mengistu Hailemariam Zenebe 

Academic Editor

PLOS ONE